# Two New ^β^*N*-Alkanoyl-5-Hydroxytryptamides with Relevant Antinociceptive Activity

**DOI:** 10.3390/biomedicines9050455

**Published:** 2021-04-22

**Authors:** Jorge Luis Amorim, Fernanda Alves Lima, Ana Laura Macedo Brand, Silvio Cunha, Claudia Moraes Rezende, Patricia Dias Fernandes

**Affiliations:** 1Laboratório de Farmacologia da Dor e da Inflamação, Instituto de Ciências Biomédicas, Universidade Federal do Rio de Janeiro, Rio de Janeiro 21941-902, Brazil; jl.amorim.22@gmail.com; 2Centro de Tecnologia, Laboratório de Análise de Aromas, Instituto de Química, Universidade Federal do Rio de Janeiro, Rio de Janeiro 21941-909, Brazil; fernanda_lima@id.uff.br (F.A.L.); alaurambrand@gmail.com (A.L.M.B.); claudia.rezendeufrj@gmail.com (C.M.R.); 3Grupo de Pesquisa em Síntese Química e Bioatividade Molecular, Instituto de Química, Universidade Federal da Bahia, Salvador 40170-115, Brazil; silviodc@ufba.br

**Keywords:** serotonin amides, ^β^*N*-alkanoyl-5-hydroxytryptamides, antinociceptive activity, pain, mechanochemical synthesis

## Abstract

In this work, we describe a new route for the synthesis and the antinociceptive effects of two new ^β^*N*-alkanoyl-5-hydroxytryptamides (named C_20:0_-5HT and C_22:0_-5HT). The antinociceptive activities were evaluated using well-known models of thermal-induced (reaction to a heated plate, the hot plate model) or chemical-induced (licking response to paw injection of formalin, capsaicin, or glutamate) nociception. The mechanism of action for C_20:0_-5HT and C_22:0_-5HT was evaluated using naloxone (opioid receptor antagonist, 1 mg/kg), atropine (muscarinic receptor antagonist, 1 mg/kg), AM251 (cannabinoid CB1 receptor antagonist, 1 mg/kg), or ondansetron (5-HT3 serotoninergic receptor antagonist, 0.5 mg/kg) 30 min prior to C_20:0_-5HT or C_22:0_-5HT. The substances both presented significant effects by reducing licking behavior induced by formalin, capsaicin, and glutamate and increasing the latency time in the hot plate model. Opioidergic, muscarinic, cannabinoid, and serotoninergic pathways seem to be involved in the antinociceptive activity since their antagonists reversed the observed effect. Opioid receptors are partially involved due to tolerant mice demonstrating less antinociception when treated with both compounds. Our data showed a quicker and simpler route for the synthesis of the new ^β^*N*-alkanoyl-5-hydroxytryptamides. Both compounds demonstrated significant antinociceptive effects. These new compounds could be used as a scaffold for the synthesis of analogues with promising antinociceptive effects.

## 1. Introduction

As recently revised by the International Association for the Study of Pain (IASP), pain is an unpleasant sensory and emotional experience associated with, or resembling that associated with, actual or potential tissue damage [1] causing high health costs and leading to economic loss to society. Available analgesics are not adequate to meet the clinical needs, leaving an increasing number of people with undertreated pain. Opioids remain the most effective analgesics for many painful conditions. However, severe undesired side effects (i.e., constipation, sedation, tolerance, and dependence) limit clinical opioid use in such a way that their use is widely accepted in cases of severe pain conditions related to cancer or end of life, and its use in chronic opioid therapy remains controversial [2].

Another pharmacological alternative for pain relief is the use of non-steroidal anti-inflammatory drugs (NSAIDs). They are important for their analgesic and anti-inflammatory properties and have been widely used for the symptomatic treatment of acute pain, and the mechanism of action is due to the inhibition of cyclooxygenase enzymes [3]. However, NSAIDs also present undesirable side effects, which ultimately limit their use [4].

Coffee is among the most popular drinks in the world, enjoyed by most cultures [5]. The waxy surface of green coffee beans consists of a soluble phase and an insoluble one comprised basically of ^β^*N*-alkanoyl-5-hydroxytryptamides (C_n_-5HT) [6,7].

Simple amides of fatty acids present anti-inflammatory or immunomodulatory and potential antitumor effects [7]. Serotonin amides have been shown to play an important role in the regulation of gastric acid secretion and have presented an anti-inflammatory effect by inhibiting the expression of caspases, mediators of arachidonic acid, and by being an antagonist in the transient receptor potential cation channel subfamily V member 1 (TRPV1) system [8,9,10,11,12,13].

The strategies that have been reported so far for the synthesis of C_n_-5HT have several limitations, such as low yield, two or three consecutive and laborious steps, and the use of toxic reagents and high-boiling solvents, hence, the need for a new synthetic strategy to produce these molecules is urgent [14,15]. Mechanochemistry is a powerful tool for molecular synthesis, in which the energy required for the reaction to occur is induced by the absorption of mechanical energy from the shock between the spheres in a ball mill. Performed in a single step and in the absence of solvents, the mechanochemistry-based strategies are rapid with high selectivity and reduced reaction times, offering evident advantages over solution-based reactions [16].

Thus, to corroborate with the investigation in serotonin amides, in this paper, we describe the one-pot mechanosynthesis and the antinociceptive activity of ^β^*N*-arachinoyl-5-hydroxytryptamide (C_20:0_-5HT) and ^β^*N*-behenoyl-5-hydroxytryptamide (C_22:0_-5HT) and suggest their mechanism of action.

## 2. Materials and Methods

### 2.1. Reagents and Laboratories Supplies

Acetylsalicylic acid (ASA), atropine sulphate monohydrate, capsaicin, L-glutamic acid (glutamate), AM251 (*N*-(piperidin-1-yl)-(4-iodophenyl)-1-(2,4-dichlorophenyl)-4-methyl-1H-pyrazole-3-carboxamide), ondansetron hydrochloride dehydrate, serotonin hydrochloride, arachidic acid, behenic acid, *N*-(3-dimethylaminopropyl)-N′-ethylcarbodiimide hydrochloride 98% (EDC.HCl), 4-dimethylaminopyridine (DMAP), capsazepine, CNQX (6-cyano-7-nitroquinoxaline-2,3-dione disodium salt hydrate), sodium chloride PA, and citric acid 99% were purchased from Sigma-Aldrich (St. Louis, MO, USA). Ethanol and formalin were purchased from Merck Inc. (Rio de Janeiro City, Rio de Janeiro, Brazil). Morphine sulphate and naloxone hydrochloride were kindly provided by Cristália (São Paulo, Brazil). Chloroform 99% and tert-butyl methyl ether 99% (TBME) were purchased from Merck Inc. (Rio de Janeiro City, Rio de Janeiro, Brazil). Drugs were dissolved in phosphate buffer saline (PBS) prior to use. Capsaicin was dissolved in 80% (*v*/*v*) ethanol plus 20% PBS (total amount of ethanol injected in paw was 40%). All drugs were diluted just before their use.

### 2.2. Animals

Swiss Webster mice (20–25 g) were donated by the Instituto Vital Brazil (Niteroi, Rio de Janeiro, Brazil). A total of 506 animals were used in this study. The animals were maintained in a room with a light-dark cycle of 12 h, 22 ± 2 °C, 60% to 80% humidity, and with food and water provided ad libitum. Animals were acclimatized to the laboratory conditions for at least 1 h before each test onset and were used only once throughout the experiments. All protocols were conducted in accordance with the Guidelines on Ethical Standards for Investigation of Experimental Pain in Animals [17] and followed the principles and guidelines adopted by the National Council for the Control of Animal Experimentation (CONCEA), approved by the Ethical Committee for Animal Research (CEUA, approval in April/30/2019 and June/18/2019, receiving the numbers no. 31/19 and 34/19, respectively). All experimental protocols were performed during the light phase. Animal numbers per group were kept to a minimum and at the end of each experiment the mice were euthanized by ketamine/xylazine overdose.

### 2.3. Synthesis of ^β^N-Alkanoyl-5-Hydroxytryptamides

Reaction progress was monitored by thin-layer chromatography (TLC) performed on aluminum sheets precoated with silica gel G 60 F254 (Merck, São Paulo, Brazil). Melting points were recorded in two repeated trials on a capillary melting point apparatus (uncorrected); NMR spectra were obtained in DMSO-d6 (Merck, Darmstadt, Germany) on a Varian VNMRS 500 MHz spectrometer. Chemical shifts (δ) are reported in ppm from TMS as the internal standard and coupling constants (J) are expressed in Hertz. Exact masses were obtained in a Dionex UltiMate 3000 liquid chromatography coupled to a hybrid Quadrupole-Orbitrap high-resolution mass spectrometer (Thermo Q-Exactive Plus, Thermo Fisher Scientific, Waltham, MA, USA) equipped with an electrospray ionization (ESI) source. Chromatography separation was performed in a Syncronis C-18 column (50 mm × 2.1 mm × 1.7 µm) in gradient elution mode using water as mobile phase A and methanol as mobile phase B both with 0.1% formic acid and 5 mM ammonium formate as follows: 0–4 min 5–30% B, 4.1–10.0 min 30–50% B, 10–14 min 50–98% B, 14–17 min 98% B, and 17.1–22 min 5% B. The column temperature was set to 40 °C and the solvent flow rate was 0.350 mL/min. Sample injection was 8 µL and samples were analyzed in positive ionization modes. Mass spectrometry conditions included spray voltage 3.9 kV (ESI+); ion transfer capillary 300 °C; sheath and auxiliary gases 50 and 15 arbitrary units, respectively; and normalized collision energy (NCE) of 22 for positive mode. Data were acquired in PRM experiment at a resolution of 35,000, content an inclusion list with C_20:0_-5HT and C_22:0_-5HT theoretical exact mass.

### 2.4. General Procedure for the Synthesis of ^β^N-Arachinoyl-5-Hydroxytryptamide (C_20:0_-5HT) and ^β^N-Behenoyl-5-Hydroxytryptamide (C_22:0_-5HT) Using a Planetary Ball Mill

Serotonin hydrochloride (74.4 mg, 0.35 mmol), EDC.HCl (67.1 mg, 0.35 mmol), DMAP (85.5 mg, 0.70 mmol), sodium chloride (409.0 mg, 7.00 mmol), and corresponding fatty acid (0.38 mmol) were added to a 50 mL stainless steel jar containing three 10 mm balls of the same material. The jar was coupled to a planetary ball mill (Retsch PM-100, Retsch GmbH & Co. KG, Haan, Germany) and agitated with a rotation speed of 500 rpm for 1 h. The reaction was monitored via TLC and the detection was made with iodine vapor. After the end of the reaction, the solid material was washed with distilled water, filtrated, and suspended in chloroform. The organic layer was washed successively with an aqueous solution of citric acid 1% *w/v* until total consumption of DMAP excess and serotonin. The organic layer was dried with anhydrous Na_2_SO_4_ and the solvent was evaporated under reduced pressure. In the end, the *N*-acylserotonins were purified by centrifugation with tert-butyl methyl ether (Figure 1).

### 2.5. ^β^N-Arachinoyl-5-Hydroxytryptamide (C_20:0_-5HT) (I)

White solid; 59% yield (97.15 mg); mp 109.5–110.0 °C; ^1^H NMR (500 MHz, DMSO-*d6*) *δ* 0.85 (3H, t, *J* = 6.0 Hz, CH_3_), 1.23 (32H, s, H3′–H18′), 1.48 (2H, m, H2′), 2.04 (2H, t, *J* = 7.0 Hz, H1′), 2.69 (2H, t, *J* = 7.0 Hz, H8), 3.26 (2H, m, H9), 6.57 (1H, m, H6), 6.81 (1H, s, H4), 7.00 (1H, s, H2), 7.10 (1H, m, H7), 7.83 (1H, m, NH-CO), 8.57 (1H, s, OH), 10.45 (1H, s, NH) ppm; ^13^C NMR (500 MHz, DMSO-*d6*) *δ* 13.92 (CH_3_), 22.06 (C3′–C18′), 25.29 (C2′), 25.41 (C8), 28.67 (C3′–C18′), 28.78 (C3′–C18′), 28.92 (C3′–C18′), 28.99 (C3′–C18′), 31.26 (C3′–C18′), 35.49 (C1′), 39.27 (C9), 102.22 (C4), 110.91 (C3), 111.22 (C6), 111.58 (C7), 122.96 (C2), 127.89 (C3a), 130.80 (C7a), 150.14 (C5), 171.96 (NH-CO) ppm. HRMS (ESI) for C_30_H_50_N_2_O_2_ [M+H]^+^ Exact Mass: 471.39450, found 471.39451.

### 2.6. ^β^N-Behenoyl-5-Hydroxytryptamide (C_22:0_-5HT) (II)

White solid; 64% yield (111.67 mg); mp 95.0–95.5 °C; ^1^H NMR (500 MHz, DMSO-*d6*) *δ* 0.85 (3H, t, *J* = 6.5 Hz, CH_3_), 1.23 (36H, s, H3′–H20′), 1.48 (2H, m, H2′), 2.04 (2H, t, *J* = 6.8 Hz, H1′), 2.69 (2H, t, *J* = 6.8 Hz, H8), 3.26 (2H, dd, *J* = 13.4 and 6.8 Hz, H9), 6.57 (1H, m, H6), 6.81 (1H, s, H4), 7.01 (1H, s, H2), 7.10 (1H, m, H7), 7.84 (1H, m, NH-CO), 8.57 (1H, s, OH), 10.46 (1H, s, NH) ppm; ^13^C NMR (500 MHz, DMSO-*d6*) *δ* 13.88 (CH_3_), 22.06 (C3′–C20′), 25.26 (C2′), 25.39 (C8), 28.64 (C3′–C20′), 28.66 (C3′–C20′), 28.77 (C3′–C20′), 28.94 (C3′–C20′), 28.97 (C3′–C20′), 31.24 (C3′–C20′), 35.47 (C1′), 39.26 (C9), 102.22 (C4), 110.90 (C3), 111.21 (C6), 111.55 (C7), 122.92 (C2), 127.88 (C3a), 130.80 (C7a), 150.13 (C5), 171.94 (NH-CO) ppm. HRMS (ESI) for C_32_H_54_N_2_O_2_ [M+H]^+^ Exact Mass: 499.42580, found 499.42511.

### 2.7. Administration of ^β^N-Arachinoyl-5-Hydroxytryptamide (C_20:0_-5HT), ^β^N-Behenoyl-5-Hydroxytryptamide (C_22:0_-5HT), and Drugs

C_20:0_-5HT and C_22:0_-5HT were dissolved in dimethylsulphoxide (DMSO) to prepare 100 mg/mL stock solutions. Before their pharmacological use, solutions were freshly prepared from each stock solution using Tween as vehicle. Doses of 0.1 to 10 mg/kg (final volume of 0.1 mL per animal) were administered by oral gavage and the final DMSO percentage did not exceed 1%. Acetylsalicylic acid (ASA, 200 mg/kg) and morphine (5 mg/kg) were used as reference drugs. The dose of ASA and morphine was chosen based on previous results obtained by our group when their DE50, i.e., the dose that caused a 50% reduction in the nociceptive effect, was calculated [18]. The control group was given vehicle (tween) only.

### 2.8. Formalin-Induced Licking Behavior

This assay was performed, as described by [19] and adapted by [20]. This model is characterized by a response that occurred in two phases. The first phase (acute neurogenic pain) occurred during the first 5 min after the intraplantar injection of formalin and the second phase (inflammatory pain) occurred during the 15 to 30 min post-injection. Animals (*n* = 7, per group) received 20 μL of formalin (2.5% *v*/*v*) into the dorsal surface of the left hind paw. The time that the animal spent licking the injected paw was immediately recorded. Mice were pretreated with oral doses of C_20:0_-5HT, C_22:0_-5HT, morphine, ASA, or vehicle, 60 min before the administration of formalin.

### 2.9. Capsaicin- or Glutamate-Induced Nociception

These experimental protocols were performed, as described by Sakurada et al. [21] for capsaicin and by Beirith et al. [22] for glutamate-induced licking. Mice (*n* = 7 per group) received an intraplantar injection of capsaicin (20 μL, 1.6 μg/paw) or glutamate (20 μL, 3.7 ng/paw). Immediately after the injection, the animals were placed individually in a transparent box and the time that the animal kept licking or biting the capsaicin- or glutamate-injected paw was recorded for 5 min (capsaicin) or 15 min (glutamate) and was considered to be the nociceptive reaction. The animals were pretreated 60 min before the intraplantar injection of capsaicin or glutamate with C_20:0_-5HT, C_22:0_-5HT (0.1–10 mg/kg, p.o.), capsazepine (capsaicin receptor antagonist, 3.2 μg/paw), cyanquixaline (CNQX, glutamate receptor antagonist, 30 ng/paw), or vehicle (p.o.).

### 2.10. Hot Plate Test

Mice were tested, according to the method described by Sahley and Berntson [23] and adapted by Matheus et al. [24]. Mice (*n* = 8 per group) were placed on a hot plate (Insight Equipment, São Paulo, Brazil) set at 55 ± 1 °C. The reaction time (licking of paws or jumping) was recorded every 30 min post oral administration of C_20:0_-5HT, C_22:0_-5HT, vehicle, or morphine until 180 min. The average reaction time (in seconds) obtained at 60 and 30 min before oral administration was considered to be baseline (normal reaction to the temperature). The area under the curve (AUC) graphs were calculated from time course graphs. The following formula, which is based on the trapezoid rule, was used to calculate the AUC: AUC = 30 × IB ((min 30) + (min 60) + (min 180)/2), where IB is the increase from the baseline (in %).

### 2.11. Carrageenan-Induced Thermal Hyperalgesia

This method was performed, as described by Sammons et al. [25]. Two measures (with 30 min intervals) prior to carrageenan injection were done and the average result was referred to as “baseline”. Animals (*n* = 8 per group) received an intraplantar injection of carrageenan (25 µL/paw, 2%) and after 1 h were placed in a hot plate (Insight Equipment, São Paulo, Brazil) set at 55 ± 1 °C. The reaction to the hot plate was recorded as mentioned previously. Vehicle, C_20:0_-5HT, or C_22:0_-5HT was orally administered to mice 30 min before the injection of carrageenan. Reductions in the reaction time on the hot plate were taken as indicative of the development of inflammatory hyperalgesia.

### 2.12. Performance of ^β^N-Arachinoyl-5-Hydroxytryptamide (C_20:0_-5HT) and ^β^N-Behenoyl-5-Hydroxytryptamide (C_22:0_-5HT) after the Development of Tolerance with Morphine

This method was performed, according to Habib-Asl [26]. Mice were divided into four groups (*n* = 7 per group) as follows: (1) daily injection of saline, (2) daily injection of morphine (50 mg/kg, i.p.)+ saline on the 5th day, (3) daily injection of morphine for five consecutive days, (4) daily injection of morphine (50 mg/kg, i.p.)+ C_20:0_-5HT (10 mg/kg, p.o.) on the 5th day, and (5) and daily injection of morphine (50 mg/kg, i.p.)+ C_22:0_-5HT (10 mg/kg, p.o.) on the 5th day. After 30 min animals were evaluated in the hot plate apparatus.

### 2.13. Analysis of the Mechanisms of Action of ^β^N-Arachinoyl-5-Hydroxytryptamide (C_20:0_-5HT) and ^β^N-Behenoyl-5-Hydroxytryptamide (C_22:0_-5HT)

One of the following treatments was given i.p. 30 min before C_20:0_-5HT or C_22:0_-5HT (10 mg/kg, p.o.): naloxone (opioid receptor antagonist, 1 mg/kg), atropine (muscarinic receptor antagonist, 1 mg/kg), AM251 (cannabinoid CB1 receptor antagonist, 1 mg/kg), or ondansetron (5-HT3 serotoninergic receptor antagonist, 0.5 mg/kg). The antinociceptive effect was evaluated in the hot plate test as described above.

### 2.14. Analysis of Spontaneous Activity, Locomotor Performance, and Catalepsy

To evaluate these effects, mice (*n* = 7 per group) were previously treated with vehicle, C_20:0_-5HT, or C_22:0_-5HT (10 mg/kg). The open field method was used to evaluate the spontaneous activity of mice [27]. After oral administration of the tested substances, mice were individually placed in a box, in which the floor had marked squares (5 × 5 cm). Over a period of five minutes, the number of squares by which each mouse crossed was counted. The locomotor performance, as described by Godoy et al. [28], from 0.5 up to 3.5 h after administration was evaluated by the use of the rotarod apparatus. The number of falls from the apparatus was recorded. To evaluate a possible cataleptic condition, the experiment was performed as described by Correia et al. [29]. The animals were placed on a glass rod, supported only by the front paws at 0.25, 0.5, 1, and 1.5 h after oral treatments with the tested compounds. All animals were observed for 30 s at each time and the latency time for withdrawal of one or both paws from the bar was evaluated.

### 2.15. Statistical Analysis

The number of animals per group was indicated in each experiment. The results are presented as mean ± SD calculated using Prism Software 8.2.1 (GraphPad Software, La Jolla, CA, USA). One-way or two-way analysis of variance (ANOVA) followed by Tukey’s post hoc test was used for unpaired data when more than two groups were compared to the same control. The post hoc tests were run only if F achieved the necessary level of statistical significance. When *p* was lower than 0.05, group differences were considered significant.

## 3. Results

### 3.1. Mechanosynthesis of ^β^N-Alkanoyl-5-Hydroxytryptamides

Compounds I and II were synthesized by mechanochemical amidation reactions, adapted from Strukil et al. [30]. The reagents were added to the reaction jar and then coupled to a planetary ball mill. The reaction was performed by neat grinding (NG) using three 10 mm grinding balls, and the rotation speed was set to 500 rpm. Both rotation speed and number of spheres were chosen according to the size of the reaction vessel used (50 mL) [31]. ^β^*N*-arachinoyl-5-hydroxytryptamide (C_20:0_-5HT) and ^β^*N*-behenoyl-5-hydroxytryptamide (C_22:0_-5HT) were obtained in a moderate yield (59 and 64%, respectively) after 1 h. To investigate whether the increase in reaction time would influence the final yield, the same reaction was left for 4 h. It was possible to observe that an increase in reaction time did not significantly influence the yield (53 and 65%, respectively).

### 3.2. ^β^N-Alkanoyl-5-Hydroxytryptamides Demonstrated Antinociceptive Effect

Figure 2 and Table 1 show that C_20:0_-5HT and C_22:0_-5HT significantly inhibit both phases (first and second phases) of formalin-induced licking (C_20:0_-5HT F13,67 = 65.16, *p* < 0.001 and C_22:0_-5HT F4,23 = 9, *p* = 0.001). The three highest doses (1, 3 and 10 mg/kg) of C_20:0_-5HT reduced the licking time by 37.1% (*p* > 0.99), 52.8% (*p* = 0.0023), 66.9% (*p* = 0.0030), respectively, while C_22:0_-5HT (at 10 mg/kg) reduced the licking time by 66.9% (*p* = 0.001) as compared with the vehicle group in the first phase. In the second phase, both the two higher doses of serotonin amides (3 and 10 mg/kg) reduced the licking time by 41.2% (*p* < 0.001), 59.7% (*p* < 0.001), and 50.1% (*p* = 0.002), 50.8% (*p* < 0.001) to C_20:0_-5HT and C_22:0_-5HT as compared with the control group, respectively.

### 3.3. ^β^N-Alkanoyl-5-Hydroxytryptamides both Presented Antinociceptive Effect in the Hot Plate Model

Figure 3 shows that, even at 30 min post-oral administration, C_20:0_-5HT (10 mg/kg dose) presented a significant antinociceptive effect (F5,30 = 6.229, *p* < 0.0004). The effect lasted from 30 to 150 min (30 min, *p* = 0.003; 60 min, *p* = 0.0251; 90 min, *p* = 0.00344; 120 min, *p* = 0.0032; 150 min, *p* = 0.0045), while C_22:0_-5HT (3 mg/kg dose) had a significant effect between 60 and 90 min (60 min, *p* = 0.041; 90 min, *p* = 0.046) or 10 mg/kg dose from 60 to 120 min (60 min, *p* = 0.038; 90 min, *p* = 0.009; 120 min, *p* = 0.049). Comparing the area under the curve values, it can be noted that the higher dose of C_20:0_-5HT (10 mg/kg) presented a significant effect as compared with morphine-treated group (F9,40 = 41.65, *p* < 0.0001).

### 3.4. ^β^N-Alkanoyl-5-Hydroxytryptamides Presented Effect in the Capsaicin- and Glutamate-Induced Nociception

Pretreatment of animals with C_20:0_-5HT or C_22:0_-5HT reduced the licking behavior induced by capsaicin with the two highest doses (F9,43 = 18.46, *p* < 0.0001) as compared with the vehicle-treated groups. Reductions observed with both doses varied between 49% and 65%. However, only the 10 mg/kg dose presented a significant result when glutamate was used as an analgesic agonist (C_20:0_-5HT, *p* = 0.0228 and C_22:0_-5HT, *p* = 0.0185 as compared with vehicle-treated group). In this protocol, inhibition in the glutamate-induced licking behavior was almost 50% at the highest dose (Figure 4).

### 3.5. Thermal Anti-Hyperalgesic Effect of ^β^N-Alkanoyl-5-Hydroxytryptamides

The intraplantar injection of carrageenan-induced local hyperalgesia. The hyperalgesic effect was observed from the fourth hour post-carrageenan injection reached the maximal effect in the sixth hour and was observed at the 24th and 72nd hour. As shown in Figure 5, when the animals were pretreated with C_20:0_-5HT (at 10 mg/kg) a complete reversal of the hyperalgesia occurred 1 h after the carrageenan injection and was maintained for 6 h. At the fourth and sixth hours, an anti-hyperalgesic effect was also observed with 3 mg/kg dose (F41,168 = 46.35, *p* = 0.0001). When mice were pretreated with C_22:0_-5HT an anti-hyperalgesic effect could be observed beginning at 1 h post-carrageenan intraplantar injection and lasting until the sixth hour. The pretreatment of mice with a 3 mg/kg dose of C_22:0_-5HT resulted in a significant effect observable at 2 h and that persisted until the sixth hour (F41,168 = 47.18, *p* = 0.0001).

### 3.6. Investigation of the Antinociceptive Mechanism of Action of ^β^N-Alkanoyl-5-Hydroxytryptamides

The antagonists, naloxone (1 mg/kg, i.p.), atropine (1 mg/kg, i.p.), ondansetron (0.5 mg/kg, i.p.), or AM251 (1 mg/kg, i.p.) were used to investigate the mechanism of action of both ^β^*N*-alkanoyl-5-hydroxytryptamides. As shown in Figure 6, when injected alone, none of the antagonists developed an antinociceptive effect per se (F4,19 = 0.8138, *p* = 0.5320). The pretreatment of mice with naloxone or atropine 30 min prior to C_20:0_-5HT (*p* < 0.0001) or C_22:0_-5HT (*p* < 0.0001) resulted in a complete blockage in the antinociceptive effect of C_20:0_-5HT and C_22:0_-5HT (F7,24 = 123.7, *p* < 0.0001). Whereas, when ondansetron (*p* < 0.0001) or AM251 (*p* < 0.0001) were administered, it should be noted, that there was a partially reversion in the antinociceptive effect of both ^β^*N*-alkanoyl-5-hydroxytryptamides (F7,24 = 123.7, *p* < 0.0001) (Figure 6).

### 3.7. Morphine Did Not Induce Cross-Tolerance to the Antinociceptive Effects of ^β^N-Alkanoyl-5-Hydroxytryptamides

Animals were treated for four consecutive days with morphine. Twenty-four hours after total tolerance, mice received C_20:0_-5HT or C_22:0_-5HT (10 mg/kg). Even in morphine-tolerant mice, both amides of serotonin demonstrated an antinociceptive effect. The latency time in those tolerant animals treated with C_20:0_-5HT or C_22:0_-5HT returned to values of 122.8% and 91.3%, respectively as compared with the morphine-tolerant-treated group (F15,48 = 106.1, *p* < 0.0001) (Figure 7).

### 3.8. ^β^N-Alkanoyl-5-Hydroxytryptamides Did Not Affect Spontaneous Activity, Locomotor Performance or Induced Catalepsy

None of the substances (at 10 mg/kg dose) presented any significant effect on spontaneous activity or motor performance (Table 2).

We also evaluated a possible cataleptic effect of both ^β^*N*-alkanoyl-5-hydroxytryptamides. As can be observed in Table 3, no significant effect was shown after oral treatment of mice with both substances.

## 4. Discussion

Pain is a common event present in several pathologies that interferes with the quality of life of the population. In this work, we propose a new synthetic route and the antinociceptive effects of two new serotonin amides, ^β^*N*-arachinoyl-5-hydroxytryptamide (C_20:0_-5HT) and ^β^*N*-behenoyl-5-hydroxytryptamide (C_22:0_-5HT), in different experimental models of nociception.

The first route reported on the synthesis of serotonin amides derived from arachidic and behenic acids was described [32]. The authors used several toxic reagents in their methodologies such as phosphorus pentachloride and pyridine as a solvent. Unfortunately, the authors did not report their conversion rate and yielding. Lang and Hofmann [33] reported the *N*-acylation of serotonin hydrochloride using the appropriate acyl chlorides generated in situ, using thionyl chloride as the chlorination agent and *N*,*N*-dimethylformamide as the solvent. After three consecutive and laborious steps, the amides C_20:0_-5HT and C_22:0_-5HT were produced with 57 and 52% yield, respectively. Later, Reddy and collaborators reported, for the first time, the synthesis of different *N*-acylated serotonins in a single step using 1-ethyl-3-(3-dimethylaminopropyl)carbodiimide and 1H-benzotriazole as coupling reagents and *N*,*N*-dimethylformamide as the solvent. The compounds were obtained in good yields for amides derived from short-chain saturated fatty acids (85–86%) and long-chain unsaturated acids (81–88%) [15].

The biggest challenge in the synthesis of *N*-serotonin amides derived from long-chain fatty acids is the low solubility of the starting materials in conventional solvents. The main routes described in the literature use *N*,*N*-dimethylformamide, a toxic and difficult-to-handle solvent, due to its high boiling point, which hinders the work-up process. The mechanochemical strategy has as the main advantage to this problem, the realization of the reaction in solid phase, without using any solvent. In addition, the reaction is carried out in one pot, reducing drastically the number of steps. In addition, it does not need to be performed under anhydrous conditions and is performed at room temperature, making the whole process less laborious. For the first time, the synthesis of ^β^*N*-alkanoyl-5-hydroxytryptamides was described in the absence of a solvent, obtaining the products in satisfactory yields.

Our data indicate that both substances produced an antinociceptive effect since an inhibition in the licking behavior induced by formalin was observed. This model is characterized by a biphasic antinociceptive model, with a neurogenic phase (first phase) that occurs immediately after the chemical stimulus and an inflammatory phase (second phase) occurring due to the release of various pro-inflammatory mediators [34,35]. Studies have demonstrated that Aδ- and C-fiber nociceptors are involved in both phases, while non-nociceptive Aβ fibers are activated during the first phase [36]. The inhibitory effect of C_20:0_-5HT and C_22:0_-5HT in the first phase may be due to the action in one or more stages of the peripheral nociceptive transduction process, as well as a direct action on opioid receptors or the inhibition of the release of substances such as glutamate, nitric oxide, and substance P, resulting in a reduction in signal transduction to the central nervous system (CNS). It is also possible that some inflammatory mediators may be inhibited since it is well known that serotonin and histamine can be released a few minutes after formalin injection in mice paws [37].

We also evaluated the effect of serotonin amides on the supra-spinal antinociceptive mechanism through the hot plate assay. The high temperature of the plate activates nociceptors located in the paws and the acute nociceptive information is transmitted to specific regions of the CNS, triggering a reflex response by the animal. Tested compounds showed an antinociceptive effect in the first 30 min, acting faster than morphine. The serotonin amides have a long carbonic chain, leading to high liposolubility. Their passage through the blood-brain barrier may be facilitated, acting directly on the central nervous system and triggering the antinociceptive effect, which may explain its rapid effect.

Capsaicin acts through TRPV1 receptors, which are expressed on nociceptive fibers, in unmyelinated C-fibers and thinly myelinated sensory A-fibers [38,39]. Several studies have shown the expression of TRPV1 receptors in the dorsal root ganglion of the spinal cord, trigeminal, and central nervous system [40]. Glutamate is an excitatory amino acid of great relevance in the sensitization of the dorsal horn of the spinal cord [40] that can be released after the stimulation of TRPV1 receptors from primary sensory neurons, which may also contribute to the overall nociceptive process [41]. The observed effects of both tested compounds can be considered to be indicative of their antinociceptive action that may be the result of direct modulation through the blocking of TRPV1 or NMDA receptors or that they could still be acting on other mediators integrated with these systems. Our data corroborate with the capsaicin assay since, in inflammatory pain conditions, the pathways for pain transmission comprise peripheral polymodal nociceptors sensitive to protons that depolarize sensory neurons by directly activating cationic channels such as TRPV1 [38].

Injection of carrageenan leads to activation of different pathways and release of several mediators (i.e., bradykinin, eicosanoids, histamine, and oxygen free radicals) producing inflammatory hyperalgesia [42]. Knowing that several endogenous systems are involved in pain control, the mechanism of antinociceptive action of C_20:0_-5HT and C_22:0_-5HT was investigated through administration of atropine (a muscarinic receptor antagonist). In addition to its modulatory quality, acetylcholine (Ach) also acts as one of the most prominent neurotransmitters in both the central and peripheral nervous systems. Peripherally, cholinergic neurons control the sympathetic and parasympathetic branches of the autonomous nervous system on the ganglionic level. In the CNS, Ach acts as a neurotransmitter and neuromodulator upon release from key groups of cholinergic projection and interneurons in both brain as well as spinal cord. Studies on the cholinergic pathway show that M2 muscarinic receptor activation may reduce the peripheral response of nociceptors to noxious stimulus. In the dorsal horn of the spinal cord, the stimulation of muscarinic receptors contributes to the analgesic effect by inhibiting the inhibitory interneurons of lamina II reducing nociceptive transmission [43].

Serotonin may act through several receptors inhibiting or facilitating nociceptive transmission [44]. Our results showed that both compounds had their antinociceptive effect reverted after administration of atropine and ondansetron, indicating that the tested amides could act on both nociceptive pathways, cholinergic and serotoninergic. The complexity of the interactions between the cholinergic pathway and other pain systems was demonstrated by the fact that the antinociception, resulting from the activation of cholinergic receptors distributed in a diversity of areas in the CNS such as the insular cortex, amygdala, prefrontal cortex, anterior cingulate cortex, is involved in the activation of the descending serotoninergic system located at the nucleus raphe magnus [40].

Opioid receptors are found at the peripheral, spinal, and supra-spinal levels of the central nervous system. Activation of these receptors leads to an inhibitory effect of calcium influx resulting in a decrease in the release of neurotransmitters, such as glutamate and substance P in the dorsal horn, a mechanism that is involved in the ascending control of pain [45]. Opioids may also induce several desired conditions. Tolerance is the severe toxicity and is defined as a reduction in the analgesic effect following prolonged drug administration resulting in a loss of potency [26]. Other important side effects of opioids and other analgesic drugs are sedation and loss of locomotor rigidity [46].

The cannabinoid tetrad is another important instrument for the characterization of CB1 receptor agonists. This designation is due to the four main effects of systemic treatment with cannabinoid receptor agonists, i.e., analgesia, hypolocomotion, hypothermia, and catalepsy. Our data suggest that C_20:0_-5HT and C_22:0_-5HT could be reducing opioid receptor signaling, therefore, reducing signal transduction and nociceptive activity. Besides causing antinociception, the two new compounds do not induce cross-tolerance since hypolocomotion and catalepsy were not observed.

As the pain control system is characterized by the involvement of several signaling pathways acting together, the confirmation and identification of the molecular targets of the tested compounds using only in vivo models prove to be difficult. Another limitation is the fact that drugs can cross the blood-brain barrier making it difficult to affirm the correct local of action (whether peripheral or central) for both substances. One possibility could be the use of drugs that do not reach the CNS (i.e., naloxone methiodide or methylnaltrexone). Unfortunately, the accessibility to those drugs is limited due to control by the federal police to prevent indiscriminate access. Thus, the data presented in this work cannot be considered to be conclusive, but they are rather suggestive of their possible mechanism of action. In order to prove the real mechanism of action and targets/receptors, in vivo assays using selective drugs and in vitro tests such as binding to several receptors must be performed. However, at the moment, these tests are not possible due to the low solubility of the compounds. Efforts are being made to improve the solubility of the compounds, thus, allowing in vitro tests to be carried out.

## Figures and Tables

**Figure 1 biomedicines-09-00455-f001:**
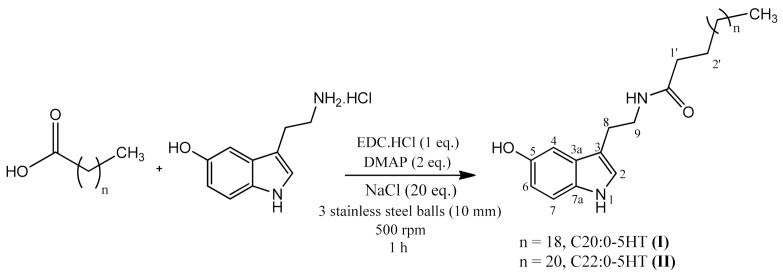
Synthesis of ^β^*N*-acylserotonins. (**I**) ^β^*N*-arachinoyl-5-hydroxytryptamide (C_20:0_-5HT); (**II**) ^β^*N*-behenoyl-5-hydroxytryptamide (C_22:0_-5HT).

**Figure 2 biomedicines-09-00455-f002:**
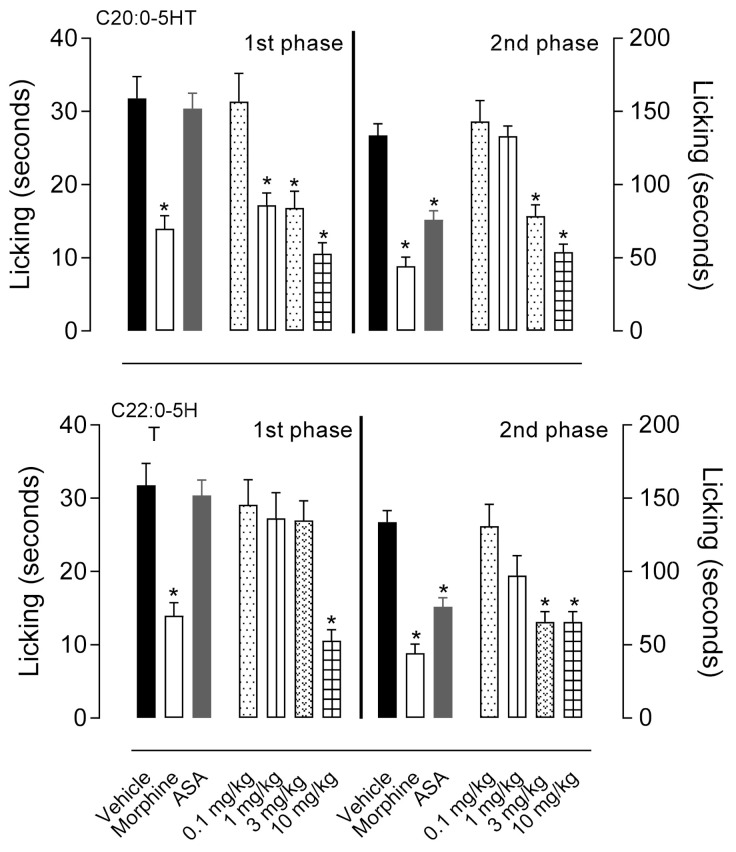
Effects of ^β^*N*-arachinoyl-5-hydroxytryptamide (C_20:0_-5HT) and ^β^*N*-behenoyl-5-hydroxytryptamide (C_22:0_-5HT) on formalin-induced licking behavior in mice. Animals were pretreated orally with C_20:0_-5HT or C_22:0_-5HT, morphine (5 mg/kg), acetylsalicylic acid (ASA, 200 mg/kg), or vehicle, and the time of licking the formalin-injected paw was recorded (in seconds) at the first phase (0–15 min post-formalin injection) and second phase (15–30 min post-formalin injection). The results are presented as mean ± SD (*n* = 7 animals per group). The average in the morphine groups was significantly decreased as compared with the respective vehicle group to each phase (first or second phase). All three doses of C_20:0_-5HT significantly reduced the licking time in the 1st phase and the two higher doses of the same compound reduced it in the 2nd phase. The dose of 10 mg/kg of C_22:0_-5HT significantly reduced the licking time in the 1st phase and the two higher doses of the compound reduced it in the 2nd phase. ASA was decreased as compared with the vehicle group in the 2nd phase only. Statistical significance was calculated by one-way ANOVA followed by Tukey’s test. * *p* < 0.05 as compared with vehicle-treated mice.

**Figure 3 biomedicines-09-00455-f003:**
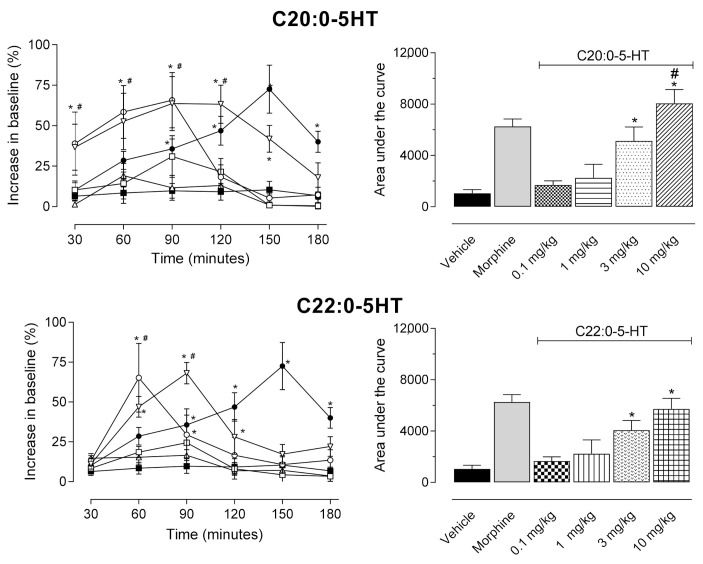
Antinociceptive effects of ^β^*N*-arachinoyl-5-hydroxytryptamide (C_20:0_-5HT) and ^β^*N*-behenoyl-5-hydroxytryptamide (C_22:0_-5HT) evaluated in the hot plate model. Animals were orally pretreated with different doses of C_20:0_-5HT or C_22:0_-5HT, morphine (5 mg/kg), or vehicle. Left graphs (line graphs) represent the time of response to the hot plate (shown as a percentage increase in relation to the baseline) evaluated between 30 and 180 min after oral administration. The right graphs represent the area under the curve calculated based on data obtained from line graphs. The results are presented as mean ± SD (*n* = 8 per group) of the increase in baseline (left graphs) or area under the curve (right graphs) calculated by two-way ANOVA followed by Tukey’s post hoc test. Morphine and higher doses (3 and 10 mg/kg) of C_20:0_-5HT and C_22:0_-5HT showed a significant increase in AUC as compared with the vehicle-treated group. C_20:0_-5HT (at 30 mg/kg) demonstrated a significant effect as compared with the morphine-treated group. * *p* < 0.05 as compared with the vehicle-treated group and # *p* < 0.05 as compared with the morphine-treated group.

**Figure 4 biomedicines-09-00455-f004:**
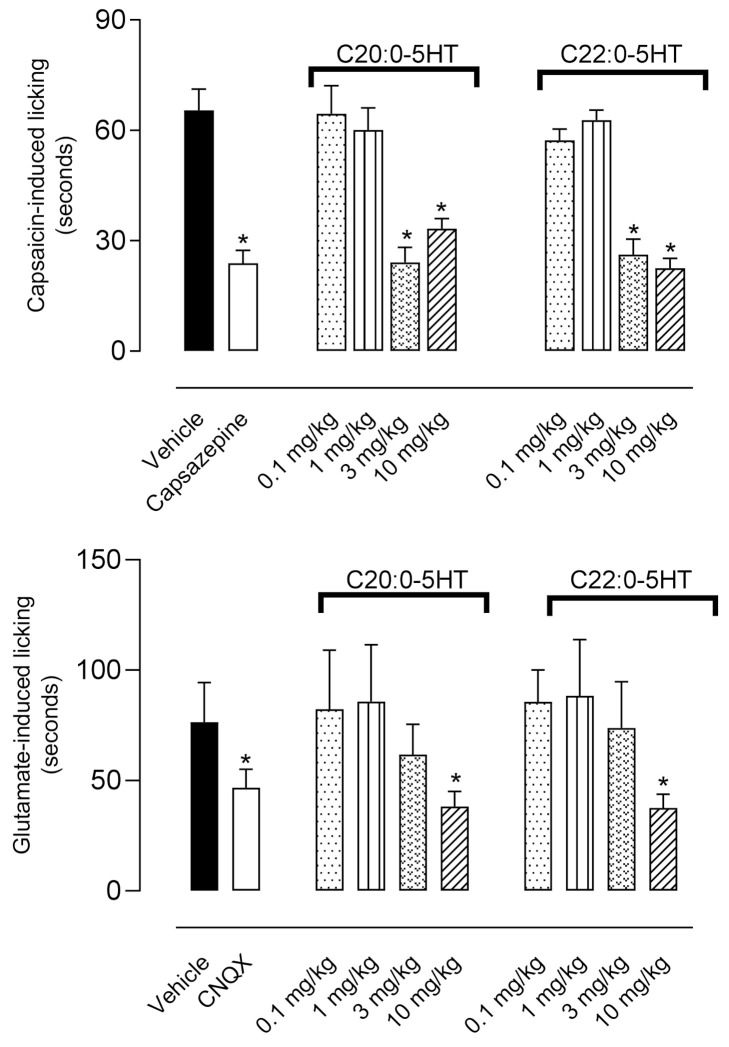
Antinociceptive effect of ^β^*N*-arachinoyl-5-hydroxytryptamide (C_20:0_-5HT) and ^β^*N*-behenoyl-5-hydroxytryptamide (C_22:0_-5HT) on the licking response induced by capsaicin or glutamate in mice. Animals were pretreated with different doses of C_20:0_-5HT, C_22:0_-5HT, or vehicle 60 min before the injection of capsaicin (1.6 µg/paw) or glutamate (3.7 ng/paw). Capsazepine (capsaicin receptor antagonist, 3.2 μg/paw) or CNQX (glutamate receptor antagonist, 30 ng/paw) were injected in mice paws 15 min prior to each respective agonist. The results are presented as mean ± SD (*n* = 7 per group) of the time that the animal spent licking the capsaicin- or glutamate-injected paw. One-way ANOVA followed by Tukey’s post hoc test was used to calculate the statistical significance. * *p* < 0.05 as compared with vehicle-treated mice.

**Figure 5 biomedicines-09-00455-f005:**
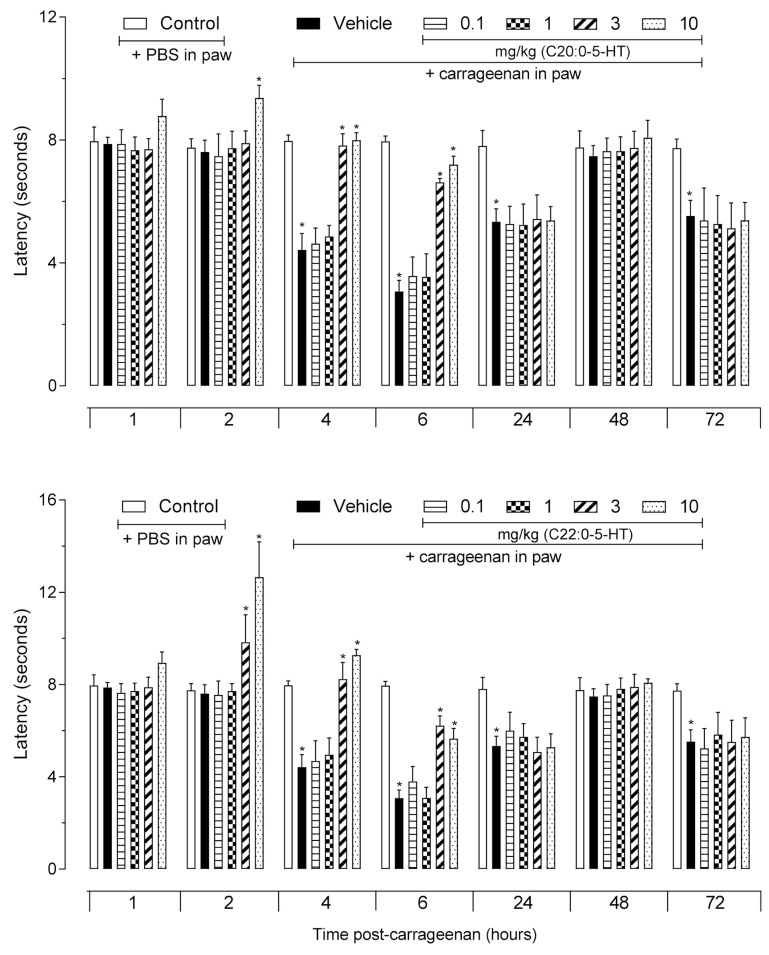
Antihyperalgesic effect of ^β^*N*-arachinoyl-5-hydroxytryptamide (C_20:0_-5HT) and ^β^*N*-behenoyl-5-hydroxytryptamide (C_22:0_-5HT). C_20:0_-5HT or C_22:0_-5HT were administered 60 min before the intraplantar injection of carrageenan (25 µL, 2%). Nociception was evaluated at each time point indicated using the hot plate system. The results are presented as mean ± SD (*n* = 8). One-way ANOVA followed by Tukey’s post hoc test was used to calculate the statistical significance. * *p* < 0.0001 comparing the C_20:0_-5HT- or C_22:0_-5HT-treated groups in mice receiving carrageenan in the paw with the vehicle-treated groups that received carrageenan in the paw.

**Figure 6 biomedicines-09-00455-f006:**
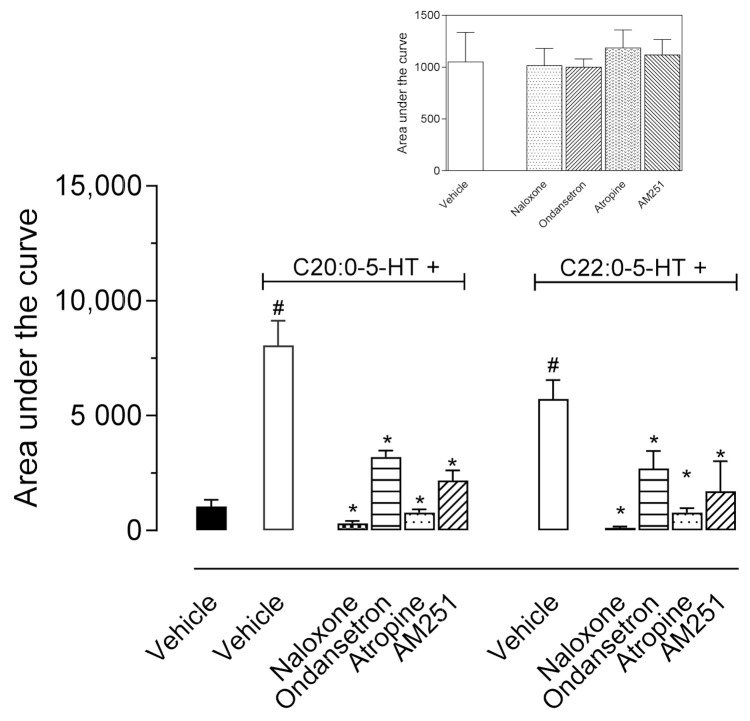
Effects of different antagonists on the antinociceptive activity of ^β^*N*-arachinoyl-5-hydroxytryptamide (C_20:0_-5HT) and ^β^*N*-behenoyl-5-hydroxytryptamide (C_22:0_-5HT) evaluated in the hot plate model. Naloxone (1 mg/kg, i.p.), ondansetron (0.5 mg/kg, i.p.), atropine (1 mg/kg, i.p.), or AM251 (1 mg/kg, i.p.), were administered 30 min prior to oral administration of C_20:0_-5HT or C_22:0_-5HT (10 mg/kg). All antagonists completely (naloxone and atropine) or partially (ondansetron and AM251) reverted the antinociceptive effect of C_20:0_-5HT and C_22:0_-5HT. Data are expressed as mean ± SD (*n* = 8). One-way ANOVA followed by Tukey’s post hoc test was used to calculate the statistical significance. # *p* < 0.05, when comparing C_20:0_-5HT- or C_22:0_-5HT-treated mice to the vehicle-treated group and * *p* < 0.05, when comparing antagonists pretreated mice with C_20:0_-5HT or C_22:0_-5HT-treated groups.

**Figure 7 biomedicines-09-00455-f007:**
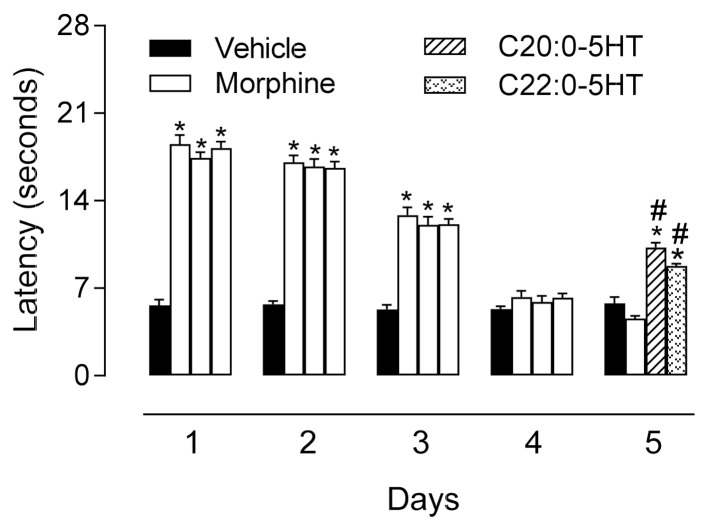
Antinociceptive effect of ^β^*N*-arachinoyl-5-hydroxytryptamide (C_20:0_-5HT) and ^β^*N*-behenoyl-5-hydroxytryptamide (C_22:0_-5HT) in morphine-tolerant mice were evaluated using a hot plate model for 5 consecutive days. Morphine (50 mg/kg, i.p.) was administrated to mice every 24 h during 4 consecutive days. On the 5th day, two tolerant groups received C_20:0_-5HT or C_22:0_-5HT (10 mg/kg, p.o.). The antinociceptive effect (indicated as the latency time to respond to the hot plate was measured in seconds) was evaluated 1 h after administration. Results are presented as mean ± SD (*n* = 7). One-way ANOVA followed by Tukey’s test was used to calculate the statistical significance. * *p* < 0.0001, when comparing treated-group with vehicle-treated groups. # *p* < 0.0001, when comparing C_20:0_-5HT- or C_22:0_-5HT-treated groups with morphine-treated group.

**Table 1 biomedicines-09-00455-t001:** Effect of ^β^*N*-alkanoyl-5-hydroxytryptamides in the formalin-induced licking response.

Group	Dose (mg/kg)	1st Phase	2nd Phase
Vehicle	-	31.8 ± 7.2	133.9 ± 18.9
Morphine	5	13.9 ± 4.3 *	44.4 ± 14.9 *
Acetylsalicylic acid	200	30.4 ± 5.1	75.9 ± 14.8 *
C_20:0_-5HT	0.1	31.4 ± 8.5	143.3 ± 31.9
	1	17.2 ± 4.1 *	133.2 ± 17.1
	3	16.8 ± 5.0 *	78.6 ± 18.6 *
	10	10.6 ± 3.6 *	53.9 ± 13.1 *
C_22:0_-5HT	0.1	29.1 ± 7.6	131.0 ± 33
	1	27.3 ± 7.7	97.3 ± 30.2
	3	27.0 ± 6.5	65.6 ± 16.0 *
	10	10.6 ± 3.6 *	65.7 ± 17.2 *

Data are presented as mean ± SD (*n* = 7). The time of licking response induced by formalin injection in the paw was recorded. Animals were pretreated orally with C_20:0_-5HT or C_22:0_-5HT, morphine (5 mg/kg), acetylsalicylic acid (ASA, 200 mg/kg), or vehicle. Statistical significance was calculated by one-way ANOVA followed by Tukey’s test. * *p* ˂ 0.05 as compared with vehicle-treated mice. -, not applicable.

**Table 2 biomedicines-09-00455-t002:** Effect of ^β^*N*-alkanoyl-5-hydroxytryptamides in spontaneous activity and locomotor performance of mice.

	Time Post Treatment (Hour)			
	0.5	1	2.5	3.5
Locomotor performance	Number of falls			
Vehicle	0.6 ± 0.5	0.6 ± 0.9	1 ± 0.7	0.8 ± 0.8
C_20:0_-5-HT	0.4 ± 0.5	0.6 ± 0.5	0.6 ± 0.5	0.4 ± 0.5
C_22:0_-5-HT	0.4 ± 0.5	0.6± 0.5	0.6 ± 0.5	0.8 ± 0.4
Spontaneous activity	Number of crossings			
Vehicle	232 ± 52.9	283 ± 64.7	204 ± 32.6	189 ± 29
C_20:0_-5-HT	278 ± 61.9	251 ± 42.7	217 ± 41.2	197 ± 31.1
C_22:0_-5-HT	254 ± 52.3	288 ± 58.8	223 ± 47.2	203 ± 34.7

Data are presented as mean ± SD (*n* = 7). It was recorded the number of falls (in the rotarod apparatus) for locomotor performance evaluation and the total number of squares (in the open field apparatus) for spontaneous activity.

**Table 3 biomedicines-09-00455-t003:** Effect of ^β^*N*-alkanoyl-5-hydroxytryptamides in the induction of a cataleptic state.

Cataleptic Effect	Number of Crossings/Minutes Post Treatment			
	15	30	60	90
Vehicle	2.93 ± 0.17	3.10 ± 0.26	4.45 ± 0.41	5.4 ± 0.75
C_20:0_-5-HT	3.25 ± 0.41	3.25 ± 0.39	4.47 ± 0.63	5.59 ± 0.49
C_22:0_-5-HT	3.22 ± 0.40	3.12 ± 0.29	4.75 ± 0.35	5.41 ± 0.54

Data are presented as mean ± SD (*n* = 7). The time (in seconds) to remove one or both paws from the bar was recorded.

## Data Availability

Not applicable

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
