# Peer review of "Two New βN-Alkanoyl-5-Hydroxytryptamides with Relevant Antinociceptive Activity"

_biomedicines, 2021, doi:10.3390/biomedicines9050455_

Round 1
Reviewer 1 Report
The present study by Jorge Luis Amorim et al. describes the synthesis and the antinociceptive effects of two novel βN-alkanoyl-5-hydroxytryptamide analogs namely C20:0-5HT and C22:0-5HT. The authors applied different acute thermal and inflammatory pain models to show the effect of the novel compounds. They also characterized in vivo the possible involvement of opioid, muscarinic, CB1, and 5-HT3 receptors. Based on these experiments the authors have stated that the novel compounds produce their antinociceptive effects by interacting with multiple receptors (opioid, muscarinic, CB1, and 5-HT3 receptor) at the periphery and CNS. My comments are addressed as follow:
Major comments
I suggest the authors to analyze the peripheral opioid analgesic effect of the novel compounds by use of systemic (sc.) naloxone or naltrexone quaternary derivative (naloxone methiodide or methylnaltrexone), that have limited CNS penetration. There are many recent data that would help the authors to improve the introduction and discussion such Balogh et al., 2018 (https://doi.org/10.1007/s11064-018-2542-7), Fürst et al., Molecules 2020, 25(11), 2473; https://doi.org/10.3390/molecules25112473 among others.
Figure 2 and 5: I ask the authors to use two-Way Repeated Measure ANOVA (there are two factors time and groups) and a correct post hoc test to compare the effects in each time points for experiments where multiple hot plate assays were used. In figure 2 the legends are missing, and in text form, the results are hardly understandable. I would recommend presenting the results in table form as well. For figure 5. I would suggest transforming the column graph to a linear type (like Figure 2). Authors need to verify why on the hot plate also 30 mg/kg dose was used in contrast to formalin, capsaicin and glutamate tests?
2.7 section: to change βN-alkanoyl-5-hydroxytryptamides did not induce tolerance to morphine did not induce cross-tolerance to the antinociceptive effects of βN-alkanoyl-5-hydroxytryptamides. The performed experiment does not inform about the tolerance inducing capacity of the test compounds, but the missing of cross tolerance after morphine treatment.
In experiments of carrageenan-induced hyperalgesia assay, the effect of both compounds was rather direct thermal antinociceptive than antihyperalgesic. The doses that caused inhibition of carrageenan-induced drop of latencies were also elevating without any drop (fig.5).
Minor comments
Introduction
Authors need to revise the definition of pain as described by IASP 2020.
Discussion
4th paragraph, typo replace presents by present (or produce)
8th paragraph. Specify the region where the cholinergic effect has proposed to be in the last sentence.
11th paragraph. use the term cross-tolerance.
Author Response
- I suggest the authors to analyze the peripheral opioid analgesic effect of the novel compounds by use of systemic (sc.) naloxone or naltrexone quaternary derivative (naloxone methiodide or methylnaltrexone), that have limited CNS penetration.
ANSWER: We would like to thanks Reviewer for this excelente suggestion. We agree that it is na excelente option to evaluate the possible mechanism. Unfortunaley, in Brazil we do not authorization to buy any substance acting in opioid system. Naloxone and morphine, used in this work, was donated by Cristália. naloxone methiodide or methylnaltrexone are sold in Sigma-Aldrich, but only to researchs outside Brazil.
- There are many recent data that would help the authors to improve the introduction and discussion such Balogh et al., 2018 (https://doi.org/10.1007/s11064-018-2542-7), Fürst et al., Molecules 2020, 25(11), 2473; https://doi.org/10.3390/molecules25112473 among others.
ANSWER: we added new phrases and information in introduction section. Please see page 1, lines 32-42 in the file Amorim et al-market text in red
In discussion we also added new phrases. Please see pages 16, lines 550-553, 565-567 in the file Amorim et al-market text in red
- Figure 2 and 5: I ask the authors to use two-Way Repeated Measure ANOVA (there are two factors time and groups) and a correct post hoc test to compare the effects in each time points for experiments where multiple hot plate assays were used.
ANSWER: We thanks the observation. We understood that Reviewer mentioned figure 3 (the hot plate model). We changed the test to two-way ANOVA. This information was added in legend. We also corrected the figure since it was missing graphs for C22.
Please see page 9, line 329 in the file Amorim et al-market text in red
- In figure 2 the legends are missing, and in text form, the results are hardly understandable. I would recommend presenting the results in table form as well.
ANSWER: We thanks the observation. Figure 2 = formalin-induced licking. Both, figure and legend, are in the text. As mentioned, we added a table with the same data.
|
Group |
Dose (mg/kg) |
1st Phase |
2nd Phase |
|
Vehicle |
- |
31.8 ± 7.2 |
133.9 ± 18.9 |
|
Morphine |
5 |
13.9 ± 4.3* |
44.4 ± 14.9* |
|
Acetylsalicilic acid |
200 |
30.4 ± 5.1 |
75.9 ± 14.8* |
|
C20:0-5HT |
0.1 |
31.4 ± 8.5 |
143.3 ± 31.9 |
|
1 |
17.2 ± 4.1* |
133.2 ± 17.1 |
|
|
3 |
16.8 ± 5.0* |
78.6 ± 18.6* |
|
|
10 |
10.6 ± 3.6* |
53.9 ± 13.1* |
|
|
C22:0-5HT |
0.1 |
29.1 ± 7.6 |
131.0 ± 33 |
|
1 |
27.3 ± 7.7 |
97.3 ± 30.2 |
|
|
3 |
27.0 ± 6.5 |
65.6 ± 16.0* |
|
|
10 |
10.6 ± 3.6* |
65.7 ± 17.2* |
Please see page 8, lines 303-308 in the file Amorim et al-market text in red
- For figure 5. I would suggest transforming the column graph to a linear type (like Figure 2). Authors need to verify why on the hot plate also 30 mg/kg dose was used in contrast to formalin, capsaicin and glutamate tests?
ANSWER: figure 5 is the hyperalgesia model. We tryed to do a line graph. However there are too many lines and graph was very confused.
We also verified the dose used and we did not find the information regarding 30 mg/kg. In all assays and in all models the higher dose used was 10 mg/kg.
- 2.7 section: to change βN-alkanoyl-5-hydroxytryptamides did not induce tolerance to morphine did not induce cross-tolerance to the antinociceptive effects of βN-alkanoyl-5-hydroxytryptamides. The performed experiment does not inform about the tolerance inducing capacity of the test compounds, but the missing of cross tolerance after morphine treatment.
ANSWER: Thank you for the suggestion. We did the modification. Please see page 13, lines 415 in the file Amorim et al-market text in red
- In experiments of carrageenan-induced hyperalgesia assay, the effect of both compounds was rather direct thermal antinociceptive than antihyperalgesic. The doses that caused inhibition of carrageenan-induced drop of latencies were also elevating without any drop (fig.5).
ANSWER: in this model, if the compound present any effect we should observe an increase in latency, suggesting an increase in the time necessary to the mouse respond to the termal stimulus. For C20 and C22 we observed a significant effect at 2,4,6 hs post- administration. It suggest that both compounds increase the time that mice need to respond to the inflammatory (carrageenan) and thermal (temperature) stimulus, indicanting an antinociceptive and/or anti-inflamatory effect.
Minor comments
Introduction
Authors need to revise the definition of pain as described by IASP 2020.
ANSWER: we changed the defition as well as the reference. Please see page 1, lines 32-35 and reference 1 in the file Amorim et al-market text in red
Discussion
4th paragraph, typo replace presents by present (or produce)
ANSWER: We changed presents to PRODUCE. Please see page 15, line 492 in the file Amorim et al-market text in red
8th paragraph. Specify the region where the cholinergic effect has proposed to be in the last sentence.
ANSWER: we added the following sentence: The complexity of the interactions between cholinergic pathway and other pain systems was demonstrated by the fact that the antinociception resulting from the activation of cholinergic receptors distributed in a diversity of areas in the central nervous system such as insular cortex, amygdala, prefrontal cortex, anterior cingulate cortex, are involved in the activation of the descending serotoninergic system located at the nucleus raphe magnus [40].
We also changed the reference 40 to a newer one.
Please see page 16, lines 550-553, and 565-567 in the file Amorim et al-market text in red
11th paragraph. use the term cross-tolerance.
ANSWER: we added the term cross-tolerance. Please see page 16, line 567 in the file Amorim et al-market text in red

Reviewer 2 Report
In the abstract section, the animal model should be better described.
In the introduction section, all acronyms should be reported in extenso at their first appearance in the text. In the line 48 there is an error in reporting the references.
In the material and method section the total number of animals should be included. In the line 89 there is an error in reporting the references.
In the figure legends, authors should include the ANOVA P value, alongside with P values related to post hoc test.
Discussion section: In the line 449 there is an error in reporting the references. Regarding the last sentence about the soulbility of these compounds for in vitro studies, should the authors refer to cell coltures, the use of DMSO as co-solvent is possible, although at percentages lower teh 1% used in the in vivo administration. I suggest a percentage lower than 0.25%.
Author Response
- In the abstract section, the animal model should be better described.
ANSWER: we added the text” The antinociceptive activities were evaluated using well-known models of thermal-induced (reaction to a heated plate, the hot plate model) or chemical-induced (licking response to paw injection of formalin, capsaicin or glutamate) nociception.” Please see page 1, lines 13-16 in the file Amorim et al-market text in red
In the introduction section, all acronyms should be reported in extenso at their first appearance in the text.
ANSWER: the missing acronym was transient receptor potential cation channel subfamily V member 1 (TRPV1). It was inserted in introduction section.
Please see page 2, line 56-57 in the file Amorim et al-market text in red
In the line 48 there is an error in reporting the references.
ANSWER: Our apologizes. this reference is wrong. It was removed from the text. Please see page 2, line 95 in the file Amorim et al-market text in red
In the material and method section the total number of animals should be included.
ANSWER: in section 2.2 it was added the total number of mice (506) used.
Please see page 2, line 90 in the file Amorim et al-market text in red
In the line 89 there is an error in reporting the references.
ANSWER: it was corrected. Please see page 2, line 96 in the file Amorim et al-market text in red
In the figure legends, authors should include the ANOVA P value, alongside with P values related to post hoc test.
ANSWER: along the text of the original submision it was included each individual p value for each group. In the legends it was included the generic indication of *. i.e., *<p.005.
Discussion section:
In the line 449 there is an error in reporting the references.
ANSWER: we did not found this error. There is any reference in line 449. However, in this sentence, we changed the place of insertion of reference 15. Please see page 15, line 510 in the file Amorim et al-market text in red
Regarding the last sentence about the soulbility of these compounds for in vitro studies, should the authors refer to cell coltures, the use of DMSO as co-solvent is possible, although at percentages lower teh 1% used in the in vivo administration. I suggest a percentage lower than 0.25%.
ANSWER: thank you for the suggestion. At the moment we are trying lower percentage of DMSO to discard a possible toxic effect of it.

Round 2
Reviewer 1 Report
The authors cannot state that both compounds demonstrated significant peripheral and central antinociceptive effects unless elucidating the involvement of the peripheral and central receptors.
The author need to analyze the peripheral opioid analgesic effect of the novel compounds by use of systemic (sc.) naloxone or naltrexone quaternary derivative (naloxone methiodide or methylnaltrexone), that have limited CNS penetration.
Also, authors can assess the peripheral involvement by administering naloxone centrally (icv.) and the test compounds orally. See Fürst et al., 2005. Peripheral versus Central Antinociceptive Actions of 6-Amino Acid-Substituted Derivatives of 14-O-Methyloxymorphone in Acute and Inflammatory Pain in the Rat. This experiment is necessary in order to support the authors' claim.
Author Response
Reviewer 1
The authors cannot state that both compounds demonstrated significant peripheral and central antinociceptive effects unless elucidating the involvement of the peripheral and central receptors.
The author need to analyze the peripheral opioid analgesic effect of the novel compounds by use of systemic (sc.) naloxone or naltrexone quaternary derivative (naloxone methiodide or methylnaltrexone), that have limited CNS penetration.
Also, authors can assess the peripheral involvement by administering naloxone centrally (icv.) and the test compounds orally. See Fürst et al., 2005. Peripheral versus Central Antinociceptive Actions of 6-Amino Acid-Substituted Derivatives of 14-O-Methyloxymorphone in Acute and Inflammatory Pain in the Rat. This experiment is necessary in order to support the authors' claim.
Answer: We’d like to thank the suggestions. Unfortunalety, in Brazil, we do not have authorization to buy naloxone methiodide or methylnaltrexone. Both drugs are controlled by Federal police. In view of this, we changed several parts of the text. We excluded several parts of the text where we stated that the antinociceptive effect could be central or peripheral in such a way that we can indicate or suggest, but we can not affirm.
We would like the reviewer to consider our explanation, the justifications and the changes made in the text so that there are no sentences where we could state the place of action (central ou peripheral).
Please see the following parts:
- Abstract
- page 1, line 19:
Original sentence: Both substances presented significant peripheral effect by reducing licking behavior induced by formalin, capsaicin, and glutamate and central antinociception by increasing the latency time in the hot plate model.
New version: Both substances presented significant effect by reducing licking behavior induced by formalin, capsaicin, and glutamate and increasing the latency time in the hot plate model.
- Line 25:
Original sentence: Both compounds demonstrated significant peripheral and central antinociceptive effects
New sentence: Both compounds demonstrated significant antinociceptive effects
- Section 3.3:
- Page 8, line 285. The title was changed to Both βN-alkanoyl-5-hydroxytryptamides presented antinociceptive effect in the hot plate model
- Page 9, line 298. The title of the legend was changed to Antinociceptive effects of βN-arachinoyl-5-hydroxytryptamide (C20:0-5HT) and βN-behenoyl-5-hydroxytryptamide (C22:0-5HT) evaluated in the hot plate model
- Discussion:
- Page 14, Line 439-440:
- Original phrase: Our data indicate that both substances produce peripheral antinociceptive effect since an inhibition in the licking behavior induced by formalin was observed
- New phrase: Our data indicate that both substances produce antinociceptive effect since an inhibition in the licking behavior induced by formalin was observed
- Page 16, Lines 516-529:
- The original paragraph: As the pain control system is characterized by the involvement of several signaling pathways acting together, the confirmation and identification of the molecular targets of the tested compounds using only in vivo models proves difficult. Thus, the data pre-sented in this work cannot be considered conclusive, but they are rather suggestive of their possible mechanism of action. In order to prove the real mechanism of action and targets/receptors, in vitro tests such as binding to several receptors, must be performed. However, at the moment these tests are not possible due to the low solubility of the compounds. Efforts are being made to improve the solubility of the compounds, thus allowing in vitro tests to be carried out.
- New paragraph: As the pain control system is characterized by the involvement of several signaling pathways acting together, the confirmation and identification of the molecular targets of the tested compounds using only in vivo models proves difficult. Another limitation is the fact that drugs can cross blood brain barrier making it difficult to affirm the correct local of action (whether peripheral or central) for both substances. One possibility could be the use of drugs that do not reach CNS (i.e., naloxone methiodide or methylnaltrexone). Unfortunately, the accessibility to thoses drugs is limited due to control by the federal polices to prevent indiscriminate access. Thus, the data presented in this work cannot be considered conclusive, but they are rather suggestive of their possible mechanism of action. In order to prove the real mechanism of action and targets/receptors, in vivo assays using selective drugs and in vitro tests such as binding to several receptors, must be performed. However, at the moment these tests are not possible due to the low solubility of the compounds. Efforts are being made to improve the solubility of the compounds, thus allowing in vitro tests to be carried out.
- Page 14, Line 439-440:

Round 3
Reviewer 1 Report
Results: Delete peripheral from the following subtitle (3.2.)
βN-alkanoyl-5-hydroxytryptamides demonstrated peripheral antinociceptive effect
Fig. 3. Make legends for the line graphs.
Author Response
Results: Delete peripheral from the following subtitle (3.2.). βN-alkanoyl-5-hydroxytryptamides demonstrated peripheral antinociceptive effect
ANSWER: Done. Please see page 6, line 254
Fig. 3. Make legends for the line graphs.
ANSWER: the original legend was “At different times (30 to 180 minutes) after oral administration the time to respond to the hot plate was measured and data was expressed as percentage increase in relation to the baseline. Line graphs were converted to area under the curve for each treated group. The results are presented as mean ± SD (n = 8 per group) of the increase in baseline or area under the curve”.
This part was changed to “Left graphs (line graphs) represents the time of response to the hot plate (showed as percentage increase in relation to the baseline) evaluated between 30 and 180 minutes after oral administration. The right graphs represents the area under the curve calculated based in data obtained from line graphs. The results are presented as mean ± SD (n = 8 per group) of the increase in baseline (left graphs) or area under the curve (right graphs)”.
Please see page 9, lines 300-305
